# Effect of Ethanol Extract of Tea on the Microstructural Features and Retrogradation Characteristics of Glutinous Rice Starch

**DOI:** 10.3390/foods13071029

**Published:** 2024-03-27

**Authors:** Shanzi Cong, Jie Ji, Xinxin Zhang, Jingyi Sun, Hongji Zhao, Xiaolan Liu, Nan Hu

**Affiliations:** 1College of Food and Bioengineering, Qiqihar University, Qiqihar 161006, China01275@qqhru.edu.cn (X.L.); 2Heilongjiang Provincial Key Laboratory of Corn Deep Processing Theory and Technology, Qiqihar 161006, China; 3Engineering Research Center of Plant Food Processing Technology, Ministry of Education, Qiqihar 161006, China

**Keywords:** glutinous rice starch, ethanol extract of tea, physicochemical properties, retrogradation properties, microstructure, in vitro digestibility

## Abstract

The glutinous rice starch (GRS) regeneration process could lead to decreased product quality and shorter shelf life. The purpose of this study was to analyze the effect of an ethanol extract of tea (EET) on the regeneration properties of GRS. The microstructure of starch was determined via scanning electron microscopy (SEM), Fourier-transform infrared (FT-IR) spectroscopy was used to determine the microstructure of starch-polyphenol molecular groups, an X-ray diffraction (XRD) instrument was used to determine the starch crystal structure, a differential scanning calorimeter (DSC) was used to determine the thermodynamic properties of starch, and the inhibitory effect of EET on GRS regeneration was comprehensively evaluated. The effect of EET on the in vitro digestion properties of GRS was also determined. The results showed that the addition of EET in GRS resulted in an increase in solubility and swelling power and a decrease in crystallinity and ΔHr. Compared to the control group, when retrograded for 10 days, the ΔHr of GRS with 1%, 2.5%, 5%, and 10% addition of EET decreased by 34.61%, 44.53%, 52.93%, and 66.79%, respectively. Furthermore, the addition of EET resulted in a decrease in the content of RDS and an increase in the content of SDS and RS in GRS. It was shown that the addition of EET could significantly inhibit the retrogradation of GRS, improve the processability, and prolong the shelf life of GRS products.

## 1. Introduction

Glutinous rice, an important cereal, occupies an important position in the dietary culture of many Asian countries [1]. It is famous for its unique viscosity and texture and is widely used in the production of various traditional foods such as zongzi, rice cakes, tangyuan, sushi, etc. [2]. The main components of glutinous rice starch (GRS) are amylose and amylopectin. GRS granules are relatively small in particle size, and the amylopectin content can reach more than 95%, while the amylose content usually does not exceed 5% [3]. Due to its good rheological and gel properties, GRS is usually used as a thickener, binder, and stabilizer [2]. However, retrogradation of rice products during storage can lead to problems such as increased hardness, difficulty in enzyme hydrolysis, and a shorter shelf life, which hinders the commercial development of glutinous rice foods [4]. Therefore, it is of great significance to effectively solve the problem of retrogradation in GRS and improve the storage stability of products.

Many techniques have been reported to inhibit starch retrograde, including physical modification [5], chemical modification [6], enzymatic treatment [7], and the addition of natural substances such as protein hydrolysates [8], plant polyphenols [9], etc. However, chemical modification typically introduces a large number of chemical reagents that may be harmful to the human body [4], while enzymatic modification can have a negative impact on the microstructure and flavor of starch granules [4]. These shortcomings limit the application of GRS in the food industry. Therefore, natural extracts have attracted the interest of researchers due to their high antioxidant properties, non-toxicity, and wide availability to inhibit starch retrogradation [7].

Tea polyphenols are one of the most significant bioactive components in tea, offering considerable health benefits [9]. Tea polyphenols have excellent biological activities, including antibacterial and antioxidant properties, and serve as non-toxic and multifunctional food additives [9]. Polyphenols in tea are water-soluble tannins, mainly including catechins, epicatechins, epicatechin gallate, epigallocatechin, and epigallocatechin gallate [10]. Black tea polyphenol extracts effectively inhibit the retrogradation of corn starch and rice starch, but they have no notable impact on the gelatinization and retrogradation characteristics of potato starch [11]. Polyphenols from green tea have been shown to prevent the aging of rice starch, corn starch, and potato starch. Furthermore, the addition of ferulic acid was found to significantly inhibit the retrograde of glutinous rice starch [12].

In addition, as a result of consumers’ attention to hyperglycemia, hyperlipidemia, and obesity, scholars in the field of food science have paid more attention to the resistant starch (RS) content in starch food. RS is defined as the total amount of starch and the products of starch degradation that resist digestion and absorption in the small intestine of the healthy human body in 120 min, but it can be fermented in the large intestine [13]. According to the different chemical structures, sources, and properties, RS can be divided into five types: RS_1_, RS_2_, RS_3_, RS_4_, and RS_5_ [14]. It provides many health benefits for humans, such as preventing diabetes, improving the intestinal microenvironment, and reducing blood sugar, blood fat, and weight [15,16]. Studies have shown that the addition of tea polyphenols to corn starch could increase the content of RS in corn starch [17].

Therefore, tea ethanolic extract was added to GRS, and the effects of EET on the physicochemical properties of GRS were analyzed by analyzing the changes in the degree of aging, the degree of solubility, the degree of swelling, and the hardness of the GRS gel. The thermodynamic properties of retrograded GRS were measured by differential scanning calorimetry (DSC), the microstructure of retrograded GRS was measured by scanning electron microscopy (SEM), and the retrogradation mechanism of GRS was characterized from a molecular perspective by infrared spectroscopy and XRD diffraction. The effect of EET on the in vitro digestion property of GRS was also analyzed. Thus, this research provided a theoretical basis for the wide application of tea polyphenols, the quality control of GRS products, and the development of related functional foods (resistant starch).

## 2. Materials and Methods

### 2.1. Materials and Reagents

Glutinous rice starch (starch content 87.16%, water content 12.28%, amylopectin content 98.23%, crude protein content 0.36%, and fat content 0.04%) and tea (tea kind: Tieguanyin) was purchased from Anxi Sijin Co., Ltd. (Quanzhou, China). Pig pancreas α-amylase (20,000 U/mL) and glucoamylase (100,000 U/g) used for in vitro digestibility were purchased from Shanghai Yuan ye Biotechnology Co., Ltd., Shanghai, China. All of the solvents used for sample preparation and extraction were purchased from Tianjin Kaitong Chemical Reagent Co., Ltd., Tianjin, China. Other reagents were obtained from Tianjin Guangfu Technology Development Co., Ltd., Tianjin, China. All reagents used were of analytical grade.

### 2.2. Preparation of EET

EET was extracted following the method of Wang et al. [17]. Tea leaves were crushed using a grinder and mixed with 70% ethanol in a 1:20 ratio (*W*:*V*). After 2 h of extraction at room temperature, the extract was centrifuged at 6000× *g,* and the residue was extracted twice using the same method. The supernatants of the three extractions were collected and transferred to a rotary evaporator and concentrated at 50 °C. The extract was then freeze-dried in a vacuum freeze dryer to obtain EET powder (yield 14.35%, total phenol content 52.31%), which was stored in a refrigerator at 4 °C.

### 2.3. Determination of Solubility and Swelling Power of GRS

The solubility and swelling power of GRS were carried out following the method of Han et al. with slight modifications [18]. Moreover, 1 g of GRS was mixed with EET powder in various proportions (0%, 1%, 2.5%, 5%, and 10%, depending on the weight of the starch) before 25 mL of distilled water was added to create a starch solution. The centrifuge tubes filled with starch paste were then incubated at 90 °C for 30 min before being cooled to room temperature. Subsequently, the solution was centrifuged, and the supernatant was placed in a Petri dish and dried at 105 °C to a constant weight. In the following, the mass of the dissolved starch was designated as A. Furthermore, the total mass of starch (1.0 g) was designated as *W*, while the mass of the residual starch paste in the centrifugal tubes was designated as P. The solubility of starch was calculated according to Formula (1). The mass of the remaining starch paste in the centrifuge tube was recorded as P, and the swelling power of the starch was calculated according to Formula (2):(1)$$Solubility(\%)=\frac{A}{W}\times 100$$
(2)$$Swelling\;power(\%)=\frac{P}{W(1-S)}\times 100$$
where A denotes the mass of the dissolved starch, *W* denotes the total mass of starch (1.0 g), and P denotes the mass of the residual starch paste in the centrifugal tubes.

### 2.4. Hardness of the Gel

The hardness of the starch gel was measured following the method of Wang et al. with slight modifications [17]. Furthermore, 0 g, 0.2 g, 0.5 g, 1.0 g, and 2.0 g EET were added to 20.0 g GRS, respectively. Then, 25 mL of distilled water was added to the GRS-EET, mixed well, and filled into a 50 mL syringe. The solution obtained was then filled into a 10 mL syringe and heated in a water bath at 90 °C to induce gelatinization. The syringe filled with starch gel was then chilled at 4 °C to form a gel column. The gel column was then poured on days 1, 3, 5, and 7 and cut into cylinders with a diameter of 1.5 cm and a height of 2 cm. Subsequently, the hardness of the gel was measured using a texture analyzer (TMS-Pro, FTC Co., Denver, CO, USA).

### 2.5. Preparation of Retrogradation Samples

A GRS mixture containing different concentrations of EET powder (0%, 1%, 2.5%, 5%, and 10%, according to starch weight) was prepared. The GRS mixture as obtained was added to 10 mL of distilled water, followed by heating in a 90 °C water bath for 30 min. After the gelatinized starch solution was cooled to 20 °C, it was stored at 4 °C for 5, 10, and 15 days for retrogradation, respectively. The retrograded starch samples were then freeze-dried (2.5 L freeze-dry system, Labconco Co., Kansas, MI, USA), ground, and sifted through a 200 mesh sieve for later use. The freeze-dried retrograded starch was used for DSC, XRD, FT-IR, SEM, and in vitro digestion analysis.

### 2.6. DSC

The thermodynamic properties of the retrograded GRS were determined using a differential scanning calorimeter (Q-20 DSC, TA Co., Boston, MA, USA) [18]. Moreover, 3.0 mg of the retrograded starch sample and 6.0 mg of deionized water were added to an aluminum crucible and compressed. The mixture of retrograded starch and water was equilibrated at room temperature for 24 h. The calorimeter was calibrated with indium standards. For all DSC runs, an empty and filled aluminum crucible was used as a control. The sample was then heated by raising the temperature. The scanning temperature range was 25 to 100 °C, and the heating rate was 10 °C/min. The measurement was carried out under ultra-high-purity nitrogen. The initial temperature (To), the peak temperature (Tp), the termination temperature (Tc), and the retrogradation enthalpy (ΔH) were recorded.

### 2.7. FT-IR Spectroscopy Properties

Retrograded GRS infrared spectroscopy analysis was performed following the method of Su et al., with minor modifications, using a Fourier-transform infrared (FT-IR) spectrometer (Spectrum-100, PerkinElmer Co., Shelton, CT, USA) [19]. Furthermore, 1.0 mg of the retrogradation samples of EET, GRS, and GRS-EET (15 d) were weighed and combined with 150 mg of KBr. The mixture was then ground together using a mortar and a pestle, and the resultant mixture was pressed using a pellet press. The FT-IR spectra of the as-obtained pellets were then recorded at a wavelength range of 4000 to 400 cm^−1^, at a scanning time of 16, and a scanning resolution of 4 cm^−1^.

### 2.8. XRD Analysis

The XRD analysis of retrograded GRS was performed following the method of Han et al., with minor modifications [20]. An X-ray diffractometer (Rint-2000, Neology Co., Tokyo, Japan) was used to scan retrograded GRS samples stored for 15 days. The scanning was performed at 20 kV and 35 mA, and the diffraction angle (2θ) was applied at a rate of 2°/min from 5° to 80°. The retrograded GRS without EET additions was used as a control. ORIGIN 8.0 (Origin Lab^®^, Northampton, MA, USA) was used to analyze the diffraction patterns.

### 2.9. SEM

The microscopic structure of GRS was determined following the method of Luo et al. with minor modifications [21]. The 15-day retrogradation GRS was freeze-dried, then ground, and sieved (200 mesh). The power samples were characterized through the use of scanning electron microscopy (SEM). The samples were affixed to aluminum stubs using double-sided adhesive carbon tape and subsequently coated with a thin layer of gold via sputter coating. The morphology of the starch granules was then observed using a ZEISS EVO18 SEM (ZEISS, Berlin, Germany), operating at an accelerating voltage of 10 kV.

### 2.10. In Vitro Digestibility

The in vitro digestibility test was performed according to the method of Englyst et al. [22]. Utilizing glucose as the standard, standard curves were generated using the following equation.
Y = 1.6134x − 0.0832 [R^2^ = 0.9991].

Subsequently, using the standard glucose curve, the amount of reducing sugar produced by starch was determined at 20 min and 120 min. The amounts of readily digestible starch (RDS), slowly digestible starch (SDS), and resistant starch (RS) in glutinous rice starch were determined using the following formulas.
(3)$$RDS\left(\%\right)=\frac{\left(G_{20}-FG\right)}{TS}\times 0.9\times 100,$$
(4)$$SDS(\%)=\frac{\left(G_{120}-G_{20}\right)}{TS}\times 0.9\times 100,$$
(5)$$RS(\%)=\frac{\left(TS-G_{120}\right)}{TS}\times 0.9\times 100.$$

In this equation, *G*_20_ denotes the glucose content produced after 20 min of enzyme hydrolysis (mg); FG denotes the content of free glucose in starch before enzyme hydrolysis (mg); TS denotes the total starch content in the sample (mg); and *G*_120_ denotes the glucose content produced after 120 min of enzyme hydrolysis (mg).

### 2.11. Statistical Analysis

Each experiment was repeated at least three times, and the results were expressed as average values with standard errors. The data were subjected to ANOVA, and a least significant difference multiple comparison test was performed using SPSS, version 11.5. *p* < 0.05 was considered statistically significant.

## 3. Results and Discussion

### 3.1. Effect of EET on the Solubility and Swelling Power of GRS

The strength of the interaction of starch with water can be judged by the magnitude of solubility and expansion [21]. Under heating conditions, starch and water molecules continuously form hydrogen bonds, and the starch begins to dissolve slightly [23]. With increasing heating time, the force within the starch molecule decreases constantly, and the undissolved starch begins to absorb water and expand, leading to an increase in the degree of expansion [24]. The effects of EET on the solubility and swelling degree of GRS are shown in Figure 1. The solubility of GRS increased with the addition of EET, and the solubility of GRS with 10% EET increased by 54.51% (*p* < 0.01) compared to the control group. This may be attributed to the fact that polyphenols can cross-link starch molecules with water molecules through the polyphenol hydroxyl, prompting the massive precipitation of amylose and amylopectin, resulting in the increased solubility of GRS [25].

The swelling power of starch in aqueous solution reflects the water-holding performance of starch [21]. The swelling power of GRS increased with the addition of EET, and the swelling power of GRS added with 10% EET increased by 41.53% (*p* < 0.01) compared to the control. The hydroxyl group of tea polyphenols can interact with amylose/amylopectin, destroying the interaction between starch chains, loosening the internal structure of starch, and causing an increase in swelling power [17]. It is also found that the complex formed by excessive polyphenols and starch will cover the surface of starch particles, which will hinder the contact of starch molecules and water molecules and reduce the expansion of starch [26].

### 3.2. Effect of EET on the Gel Hardness of GRS during Retrogradation

The degree of starch degradation increases during the storage of starch foods, leading to increased hardness and poor taste of the product [19]. This is attributed to the gelatinization of starch after cooling, where starch molecules entangle and bind together to form a three-dimensional network structure, increasing the hardness of starch-based products [27]. Therefore, hardness is the most important sensory characteristic of starchy foods during the retrogradation process and one of the most important evaluation indices for starch-based foods [28].

Figure 2 shows the effect of different amounts of EET additions on the hardness of the GRS gel during the retrograde process. With an extension of the retrogradation time, the hardness of the GRS gel gradually increases. When the retrograde time is the same, the greater the amount of EET added, the lower the hardness of GRS, and a concentration-dependent relationship is presented. Many studies have found similar results. Rutin and quercetin can also lead to a decrease in the hardness of the wheat starch-form gel [29], while plant polyphenols can significantly reduce the hardness of the sweet potato starch gel and inhibit the rate of increase in the hardness of the gel during the retrogradation process [30]. This may be because plant polyphenolic compounds can form hydrogen bonds, hydrophobic bonds, and van der Waals forces with starch molecules, thus altering the interactions between starch molecules, inhibiting, to some extent, the rearrangement and crystallization processes of starch molecules, thereby reducing the hardness of starch gels [17]. It is worth noting that when the amount of EET added in glutinous rice starch is greater than 2%, the hardness of the starch gel decreases significantly, and the increase in the hardness of the starch gel in the later stage of storage is smaller. From the first day to the seventh day, the gel strength of the GRS control group increased by 58.31%, while the gel hardness of the sample group with 10% EET addition only increased by 23.62%. This indicates that with the addition of EET (mainly polyphenols) to starch, the crystallization process of GRS is inhibited, and more time is required for crystallization.

### 3.3. Effect of EET on the Thermal Properties of GRS

When gelatinized starch is stored at low temperatures, starch molecules form crystalline polymers through hydrogen bonds [31]. Retrogradation enthalpy (ΔHr) is the energy value required for the melting of these reaggregated crystals, so ΔHr can be used to represent the degree of starch retrogradation [32,33]. As can be seen in Table 1, the ΔHr of all samples increased significantly (*p* < 0.05). Furthermore, when EET was added to GRS, the ΔHr of the GRS samples was significantly reduced (*p* < 0.05). However, the greater the amount of EET added, the smaller the ΔHr of GRS, showing a significant dose effect. When retrograded for 10 days, the ΔHr of GRS with the addition of EET of 1%, 2.5%, 5%, and 10% decreased by 34.61%, 44.53%, 52.93%, and 66.79%, respectively, compared with the control group. This indicates that EET can inhibit the recrystallization process of GRS and reduce the content of crystalline starch in retrograded GRS, thereby inhibiting the degree of GRS retrogradation. Xiao et al. found that tea polyphenols can reduce the ΔHr of rice starch [11]. The hydroxyl groups in plant polyphenols can easily form hydrogen bonds with the hydroxyl groups in starch, which can inhibit the formation of starch polymer chains, thus reducing the energy required for the recrystallization of retrograded starch when it melts [19].

### 3.4. Effect of EET on FT-IR Analysis of GRS

FT-IR is a technique that uses the interaction between infrared radiation and substances to analyze the chemical composition and structure of samples [34]. By measuring the absorption or emission of infrared light by the substance, it provides information about the functional groups and chemical bonds within the molecule [35]. Therefore, changes in the internal structure of starch molecules during the retrograde process can be qualitatively analyzed using infrared spectroscopy, reflecting the offset changes in chemical bonds within starch molecules during the retrograde process [36]. Figure 3A shows the characteristics of the polyphenolic substance with the existence of a benzene ring OH stretching at 3200–3500 cm^−1^. The appearance of a strong peak around 1450–1600 cm^−1^ is due to the stretching of the benzene ring bond, and the band around 1000–1300 cm^−1^ is caused by the C-O/C-C stretching vibration [18]. Figure 3B shows the FT-IR of retrograded GRS and GRS-EET (15 days). At 3400 cm^−1^, all 15d samples showed a strong absorption peak, probably due to the intermolecular and intramolecular tensile vibration of O-H [37]. The intensity of the broad peak around 3400 cm^−1^ decreased due to the reduced number of hydroxyl groups, indicating reduced hydrogen bonding interactions between the starch molecules [17]. Interestingly, with the addition of EET, the peak gradually moved toward a lower wavelength, from 3442 cm^−1^ to 3410 cm^−1^. This implies that the interaction between and the components may lead to a shift in the absorption peak [38]. However, no additional functional group peaks were observed, suggesting that the interaction between the EET extract and the starch was limited to hydrogen bonds and that no new compounds were formed.

### 3.5. Effect of EET on the XRD Analysis of GRS

Depending on the different crystal structures of the amylopectin branch, amylopectin crystals can generally be divided into types A, B, and C. Grain starch is mainly type A, root and tuber starch and corn starch are mainly type B, and root and bean starch are type C crystals [39]. Glutinous rice starch, as a typical high amylopectin, has a structure of type A crystals [40]. The effect of EET additions on the crystallization of wheat starch after retrogradation is shown in Figure 4. The diffraction peak of GRS appears around 17.1°, indicating that it belongs to type A crystals, and the addition of EET has no effect on the crystal type of GRS. However, the addition of EET has a significant effect on the crystallinity of GRS. As the crystallinity of starch increases, the height of the corresponding XRD diffraction peak increases, and the increase in the height of the diffraction peak indicates an increase in the degree of starch retrogradation [19]. Compared to the control sample, the diffraction peak intensity corresponding to the crystal structure of GRS with the added EET is significantly weakened, and the XRD diffraction peak intensity gradually decreases with the gradual increase in EET additions. Among them, the height of the XRD diffraction peak of the GRS group with 10% EET addition is the lowest, and the crystallization inhibition effect is the best. The possible reason for this is that the polyhydroxy structures in the extract are easily soluble in water, limiting the movement of water molecules, therefore not combining well with starch molecules and inhibiting starch crystallization [17]. Wang et al. added EET to wheat starch and found that EET can inhibit the recrystallization of wheat starch [41].

### 3.6. Effect of EET on the Microstructure of GRS

In the process of starch regeneration, starch molecules could form microcrystals. The more crystals they form, the greater the water loss, resulting in more holes formed after starch freeze-drying. Therefore, the degree of starch regeneration can be indirectly characterized according to the number of spaces in the microstructure of the starch granules [41]. Through SEM scanning, the effects of EET on the resulting structure of the GRS particles can be observed from the microstructural aspect, as shown in Figure 5. The particles in the control group were polygonal and irregular in shape, and the surface showed a rough porous structure. However, the surface of the GRS particles supplemented with 10% EET is relatively smooth. This phenomenon indicates a large water loss of the regenerated GRS in the control starch particles. However, the hydroxyl group of GRS molecules with ET forms hydrogen bonds with the hydroxyl group of polyphenols that contain a large amount of the hydroxyl group in the extract, increasing the water retention capacity of starch, so it can still maintain a relatively smooth surface form after freezing-drying [21]. These results are consistent with those reported by Wang et al. [40] and Liu et al. [41].

### 3.7. In Vitro Digestibility

The digestibility of starch is a crucial metabolic reaction that determines the level of postprandial blood glucose in the human body [42]. The overconsumption of starchy foods on a long-term basis can induce hyperglycemia and even cause type II diabetes [11]. Controlling starch consumption, particularly total digestible starch consumption, has been shown to effectively delay the increase in blood sugar level and the corresponding insulin response [43]. Englyst et al. [22] classify three types of starch, rapidly digestible starch (RDS), slowly digestible starch (SDS), and resistant starch (RS), according to their biological availability, based on the varying digestion rates of the human digestive tract. RDS is the amount of reducing sugar produced by starch in the body over a period of 20 min. SDS is the amount of reducing sugar produced by starch in the body over a period of 120 min. RS refers to the amount of starch that cannot be hydrolyzed by amylase. On the contrary, resistant starch can be fermented by intestinal microorganisms, thus playing the role of dietary fiber [17].

As can be seen in Table 2, with the extension of the retrograde time of GRS, the RDS content in GRS gradually decreases, while the SDS and RS content gradually increases. This result indicates that the in vitro digestibility of starch is inhibited. This inhibiting effect is caused by interactions between polyphenols and enzymes. The abundant hydroxyl groups in polyphenols combine with amino acid residues at the active site of the enzyme, thus decreasing the enzyme activity [44]. Although interactions between polyphenols and enzymes have been believed to be the main cause of the inhibition of starch digestion, the interactions between polyphenols and starch can also affect digestion [45]. The surface of starch particles is tightly surrounded by the hydroxyl structure of polyphenols, which reduces the surface area for starch digestion by enzymes and prevents the enzyme from binding to the starch hydrolysis site. Sun et al., found that the inhibitory effect of polyphenols on starch in vitro was reflected in two aspects: not only in the fact that polyphenols can inhibit key digestive enzymes such as amylase and beta-glucosidase but also in the result of the interaction between polyphenols and starch [42]. Wang et al. [44] also discovered that the addition of a specific amount of EET to wheat starch decreased the hydrolysis rate of starch to some extent. In conclusion, the addition of EET to glutinous rice and its starch products reduces the postprandial glycemic index, thus providing a novel method for preventing diabetes.

## 4. Conclusions

EET has significant effects on the physicochemical properties, retrogradation properties, and in vitro digestibility of GRS. The retrogradation of GRS was inhibited by the addition of EET. With the addition of EET, the solubility of GRS increased gradually, the swelling degree decreased slowly, and the gel strength decreased gradually. Furthermore, the retrogradation enthalpy and relative crystallinity of GRS with EET additions also decreased. The addition of EET to GRS reduced the digestibility of GRS, decreased the range of RDS, and increased the range of RS. Therefore, the adding of an appropriate amount of EET to GRS foods not only can reduce the degree of aging and improve the sensory quality of starch products but can also inhibit the digestibility of starch and increase its industrial value.

## Figures and Tables

**Figure 1 foods-13-01029-f001:**
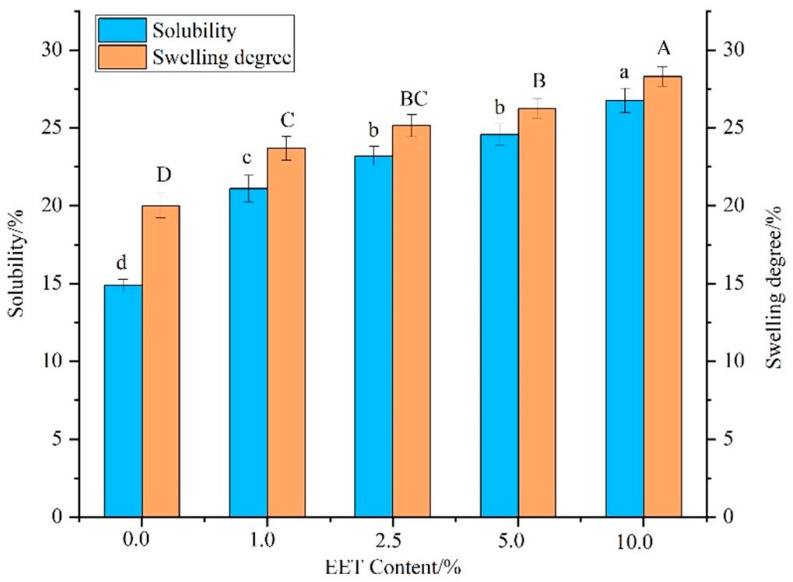
Effect of ethanol extract of tea (EET) on solubility and swelling power of glutinous rice starch (GRS). Note: different letters indicate significant differences between data (*p* < 0.05).

**Figure 2 foods-13-01029-f002:**
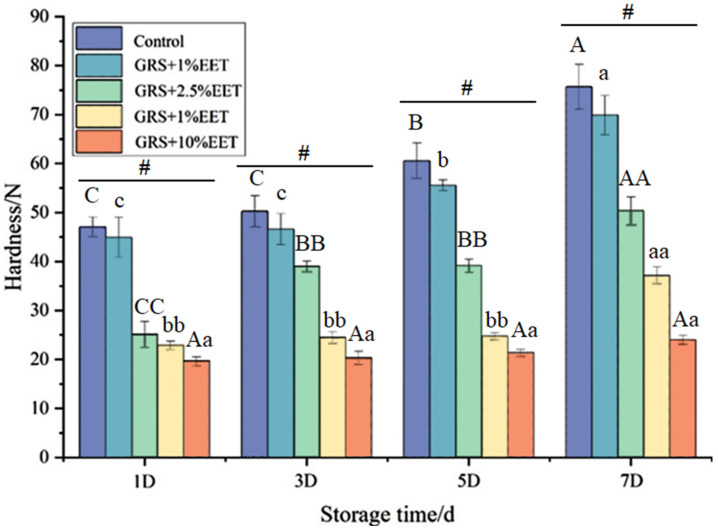
Effect of ethanol extract of tea (EET) on gel hardness of glutinous rice starch (GRS) during retrogradation. Note: different letters indicate significant differences between data (*p* < 0.05), and # indicates significant differences within groups (*p* < 0.05).

**Figure 3 foods-13-01029-f003:**
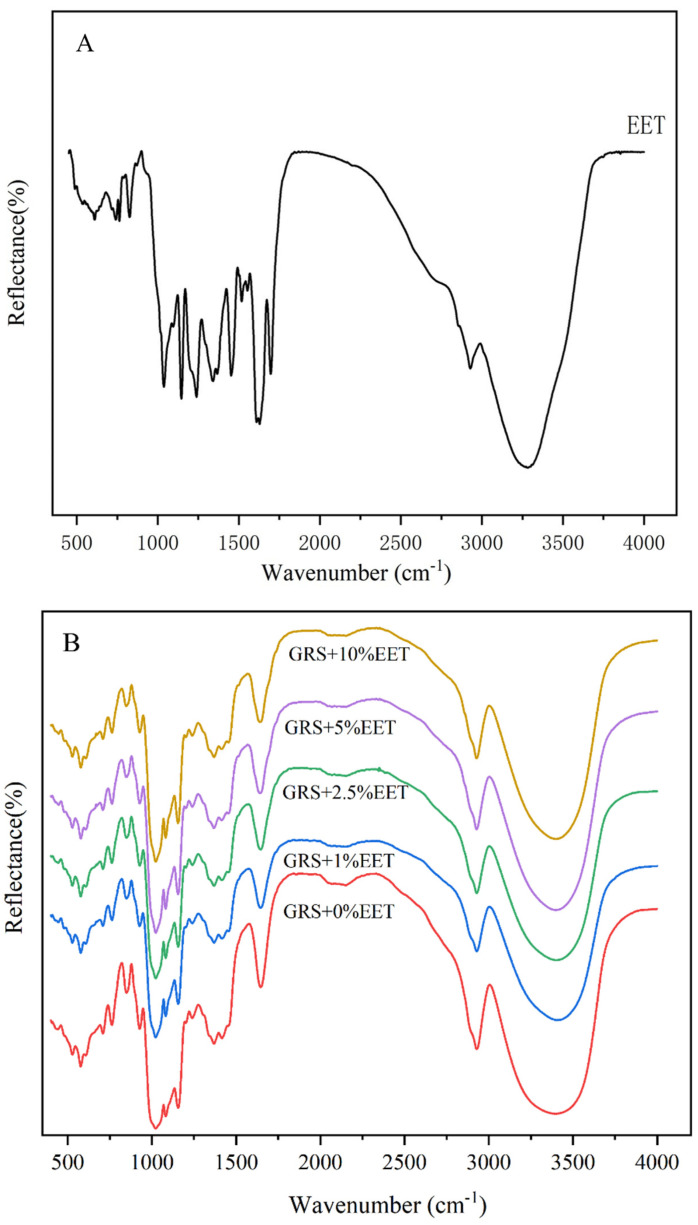
Effect of ethanol extract of tea (EET) on FT-IR spectra of glutinous rice starch (GRS) retrograded for 15 days. Note: (**A**) EET group, (**B**) GRS with different EET concentration groups added.

**Figure 4 foods-13-01029-f004:**
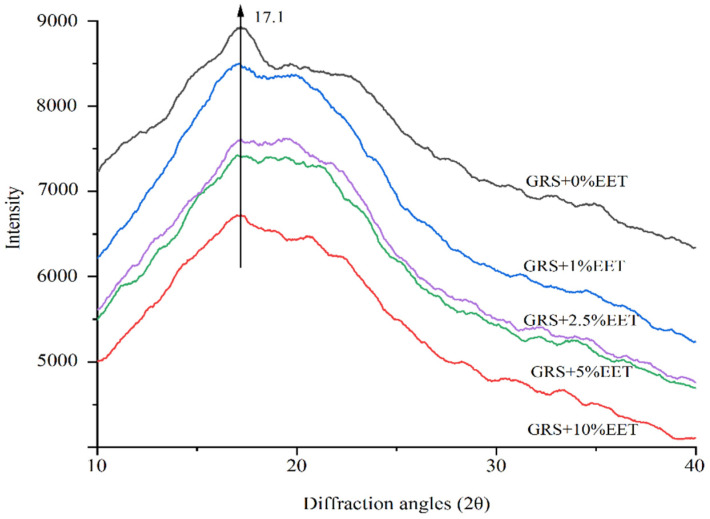
Effect of ethanol extract of tea (EET) on X-ray diffraction profile of glutinous rice starch (GRS) retrograded for 15 days.

**Figure 5 foods-13-01029-f005:**
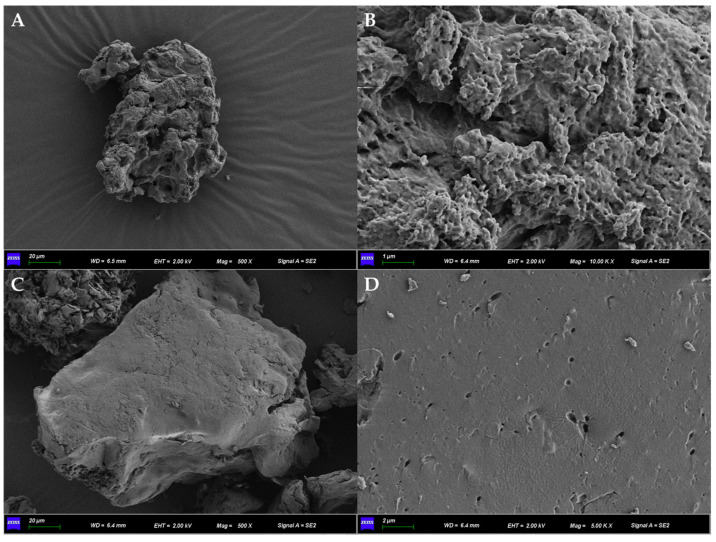
Scanning electron the micrograph images showing the influence of ethanol extract of tea (EET) on microstructure of glutinous rice starch (GRS). (**A**,**B**) control group; (**C**,**D**) added 10% EEOP extract; (**A**,**C**) 800 times the magnification; (**B**,**D**) 10,000 times the magnification.

**Table 1 foods-13-01029-t001:** Effect of ethanol extract of tea (EET) on regeneration thermodynamic properties of glutinous rice starch (GRS).

Storage	EET Content	T_0_ (°C)	T_P_ (°C)	T_C_ (°C)	ΔHg (J/g)
5 d	0%	41.48 ± 4.50 ^b^	53.5 ± 2.48 ^b^	67.58 ± 5.53 ^a^	7.46 ± 1.04 ^c^
	1%	41.37 ± 3.71 ^a^	56.04 ± 0.22 ^a^	65.50 ± 2.93 ^a^	4.46 ± 0.36 ^c^
	2.5%	40.39 ± 2.22 ^c^	52.38 ± 0.81 ^b^	64.75 ± 0.37 ^c^	3.64 ± 0.82 ^c^
	5.0%	40.10 ± 5.25 ^c^	52.65 ± 2.76 ^b^	63.33 ± 3.07 ^a^	3.07 ± 0.23 ^c^
	10.0%	40.87 ± 4.18 ^c^	53.06 ± 1.11 ^c^	59.00 ± 2.25 ^c^	1.93 ± 0.03 ^c^
10 d	0%	42.07 ± 4.36 ^a^	53.86 ± 1.04 ^a^	66.84 ± 4.32 ^b^	7.86 ± 0.72 ^b^
	1%	40.30 ± 2.87 ^b^	50.25 ± 2.24 ^c^	61.75 ± 3.16 ^a^	5.14 ± 0.26 ^b^
	2.5%	41.21 ± 0.99 ^b^	53.59 ± 1.04 ^a^	66.53 ± 3.01 ^a^	4.36 ± 0.01 ^b^
	5.0%	40.53 ± 4.25 ^b^	52.68 ± 0.27 ^a^	61.83 ± 4.62 ^b^	3.70 ± 0.35 ^b^
	10.0%	45.68 ± 3.86 ^a^	58.03 ± 1.32 ^a^	61.33 ± 3.83 ^b^	2.61 ± 0.22 ^b^
15 d	0%	42.91 ± 3.24 ^c^	51.95 ± 3.42 ^c^	66.38 ± 1.67 ^c^	10.32 ± 1.43 ^a^
	1%	41.66 ± 0.52 ^b^	50.61 ± 1.34 ^b^	65.25 ± 3.32 ^b^	7.00 ± 0.77 ^a^
	2.5%	41.26 ± 3.24 ^a^	51.38 ± 3.64 ^c^	66.25 ± 3.14 ^b^	8.35 ± 0.36 ^a^
	5.0%	42.25 ± 2.28 ^a^	52.24 ± 0.89 ^c^	65.81 ± 2.36 ^a^	6.74 ± 0.64 ^a^
	10.0%	43.11 ± 4.35 ^b^	56.30 ± 0.42 ^b^	65.50 ± 3.78 ^a^	4.80 ± 0.21 ^a^

Note: values are expressed as mean ± standard deviation (n = 3). a–c: different lower case letters in the EET content with the same concentration for different days indicate significantly different (*p* < 0.05). T_0_: initial temperature, T_P_: peak temperature, T_C_: termination temperature, and ΔH: retrogradation enthalpy.

**Table 2 foods-13-01029-t002:** Effect of ethanol extract of tea (EET) on in vitro digestive properties of retrograded glutinous rice starch (GRS).

Storage	EET Content	RDS	SDS	RS
5 d	0%	14.27 ± 1.88 ^a^	1.13 ± 0.16 ^c^	84.33 ± 3.82 ^c^
	1%	9.33 ± 4.21 ^a^	6.34 ± 1.72 ^c^	84.57 ± 4.58 ^c^
	2.5%	7.55 ± 5.82 ^a^	7.88 ± 1.43 ^c^	84.60 ± 3.28 ^c^
	5.0%	5.73 ± 2.62 ^a^	8.98 ± 1.31 ^b^	85.29 ± 3.62 ^c^
	10.0%	5.05 ± 1.28 ^a^	7.79 ± 1.43 ^c^	87.16 ± 4.95 ^b^
10 d	0%	9.18 ± 3.42 ^b^	6.15 ± 2.22 ^b^	84.67 ± 3.24 ^b^
	1%	6.90 ± 1.84 ^b^	7.88 ± 1.25 ^b^	85.22 ± 2.33 ^b^
	2.5%	5.38 ± 0.38 ^b^	8.39 ± 1.06 ^b^	86.23 ± 4.83 ^b^
	5.0%	4.53 ± 1.26 ^b^	8.58 ± 0.92 ^a^	86.60 ± 3.56 ^b^
	10.0%	3.69 ± 0.85 ^b^	9.71 ± 1.42 ^c^	86.89 ± 2.26 ^c^
15 d	0%	7.45 ± 1.82 ^c^	6.84 ± 2.68	84.74 ± 5.20 ^a^
	1%	4.23 ± 0.35 ^c^	11.03 ± 2.07 ^a^	85.71 ± 3.34 ^a^
	2.5%	3.84 ± 0.02 ^c^	8.84 ± 1.19 ^a^	87.24 ± 3.51 ^a^
	5.0%	3.32 ± 0.08 ^c^	9.44 ± 1.44 ^c^	87.32 ± 0.16 ^a^
	10.0%	3.02 ± 0.13 ^c^	9.59 ± 1.34 ^b^	87.39 ± 2.14 ^a^

Note: values are expressed as mean ± standard deviation (n = 3). a–c: different lowercase letters in the EET content with the same concentration for different days indicate significantly different (*p* < 0.05). RDS: rapidly digestible starch, SDS: slowly digestible starch, and RS: resistant starch.

## Data Availability

The original contributions presented in the study are included in the article, further inquiries can be directed to the corresponding author.

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
