# Peer review of "Effect of Ethanol Extract of Tea on the Microstructural Features and Retrogradation Characteristics of Glutinous Rice Starch"

_foods, 2024, doi:10.3390/foods13071029_

Round 1
Reviewer 1 Report
Comments and Suggestions for Authors
Below are some suggestions to make the manuscript better.
1. The manuscript needs to be revised extensively for the English language.
2. The referencing is improper in several places. Please revise the references thoroughly
3. Line 45-47, add references against each modification method and indicate what exogenous substances have been used.
4. Line 87. Please indicate the state of ‘tea’ used in this study, as there are several forms of tea available e.g., leaves or dried.
5. Please indicate the acronyms consistently throughout the manuscript e.g., GRS
6. I fail to understand that if the addition of EET causes gel firmness, then howcome the crystallinity of the starch is reduced, as shown in XRD? The same goes for the DSC results. There is a decline in enthalpy with the increasing amount of EET, irrespective of storage days. Ideally, as per your arguments, the higher amount of H-bonding should result in higher enthalpies.
7. Moreover, how come there is no additional peak detected in FTIR? Can you confirm by doing an FTIR on EET alone?
8. SEM results do not provide substantial evidences too.
Comments on the Quality of English Languagee manuscript needs to be revised extensively for the English language.
Reviewer 2 Report
Comments and Suggestions for Authors
The study reports interesting results on the properties of starch with added tea extracts. Some comments should be considered prior to the acceptance for publication.
1) The authors must clearly indicate wich tea plant they are using, as many countries have different types of tea using specific herbs for such a purpose. Simply indicating "tea" gives a vague idea.
2) Statistics in figures are missing; the authors must add lowercase letters in those graphs containing standard deviation signs, so the data can be easily evaluated by the reader. Acronyms used in figures must be explained in the legends/captions.
3) Tables are difficult to interpret without proper indication of each acronym significance in the footnote.
4) SEM images clearly shows the impact of some modifications in the starch samples, but the authors merely indicate the influence of water presence or absence. The morphological alterations must be enhanced to better explain the physical or chemical modifications.
4) XRD results must also explore the effects of tea addition on the chemical properties and include possible correlations with their chemical compounds.
5) DSC results (acronyms) in tables must be properly acknowledged in the footnote.
Comments on the Quality of English LanguagePlease double check citations formatting, there are some errors/typo.
Reviewer 3 Report
Comments and Suggestions for Authors
In the introduction, add more information about resistant starch, its properties and physiological importance.
Different type of font in the text.
Please also examine DSC in the retrogradation process and calculate %R.
The discussions of results should be improved.
Comments on the Quality of English LanguageMinor editing of English language required
Reviewer 4 Report
Comments and Suggestions for Authors
I read with interest the manuscript which examines how glutinous rice starch interacts with ethanol extract of tea at different concentrations during the gelatinization process, and the effects on various properties such as starch solubility, swelling power, gel strength, gelatinization, retrogradation, crystal structure, microstructure, and in vitro digestibility. Thus, inclusion of tea extract led to an increase in solubility and swelling power of glutinous rice starch, while decreasing gel strength. In comparison to the control group, glutinous rice starch with tea extract exhibited lower gelatinization enthalpy, retrogradation enthalpy, retrogradation rate, and relative crystallinity, indicating inhibition of retrogradation. Moreover, adding tea extract reduced the digestibility of glutinous rice starch, decreasing rapidly digestible starch content and increasing resistant starch content. In brief, incorporating an appropriate amount of tea extract in starch-based foods can help inhibit retrogradation of glutinous rice starch, enhance sensory quality, and extend shelf life.
I agree with the publication of the manuscript in Foods after certain minor changes, such as:
1. at page 2, line 89, the word "removed" should be replaced with "collected" for a better understanding of the text;
2. at page 6, line 223, the subject is missing (hydroxyl groups of hydrophilic (?!) interact);
3. all graphics should have the text used in bold (titles and axis values, legend, etc.) and instead of white and light gray as colors, more vivid colors should be used, knowing that occurrence of colors in the manuscript does not induce an additional published cost.
Round 2
Reviewer 2 Report
Comments and Suggestions for Authors
The abstract must be rewritten, as it is quite equal to that presented in doi: 10.7506/spkx1002-6630-20200212-109
Also, many references do not appear adequately in the text, and several errors showing "[Error! Reference source not found.]" were observed.
Comments on the Quality of English LanguageEnglish is fine
Author Response
Response to Reviewer 2 Comments
|
||
|
|
|
Thank you very much for taking the time to review this manuscript. Please find the detailed responses below and the corresponding revisions/corrections highlighted/in track changes in the re-submitted files.
|
||
Comments 1: The abstract must be rewritten, as it is quite equal to that presented in doi: 10.7506/spkx1002-6630-20200212-109
|
||
Response 1: Thank you for pointing this out. The abstract has been rewritten.
|
||
Comments 2: Also, many references do not appear adequately in the text, and several errors showing "[Error! Reference source not found.]" were observed.
|
||
Response 2: Thank you for pointing this out. We have made adjustments to the references. The references you did not find before may be some Chinese journals. We've replaced them this time as well.
|
Reviewer 3 Report
Comments and Suggestions for Authors
The manuscript has been corrected according to the Reviewers' suggestions.
Comments on the Quality of English LanguageMinor editing of English language required
Author Response
Response to Reviewer 3 Comments
|
||
|
|
|
Thank you very much for taking the time to review this manuscript. Please find the detailed responses below and the corresponding revisions/corrections highlighted/in track changes in the re-submitted files.
|
||
Comments 1: Minor editing of English language required.
|
||
Response 1: Thank you for pointing this out. We have touched up the language in the manuscript. |